# Promoting Healthy Aging for Older People Living with Chronic Disease by Implementing Community Health Programs: A Randomized Controlled Feasibility Study

**DOI:** 10.3390/ijerph21121667

**Published:** 2024-12-13

**Authors:** Anne-Marie Hill, Trish Starling, Wei Xin, Chiara Naseri, Dan Xu, Geraldine O’Brien, Christopher Etherton-Beer, Leon Flicker, Max Bulsara, Meg E. Morris, Sharmila Vaz

**Affiliations:** 1School of Allied Health, The University of Western Australia, Perth, WA 6009, Australia; trish.starling@uwa.edu.au (T.S.); wei.xin1@postgrad.curtin.edu.au (W.X.); chiara.naseri@sjog.org.au (C.N.); geraldine.obrien@uwa.edu.au (G.O.); sharmila.vaz@murdoch.edu.au (S.V.); 2WA Centre for Health and Ageing, The University of Western Australia, Perth, WA 6009, Australia; christopher.etherton-beer@uwa.edu.au (C.E.-B.); leon.flicker@uwa.edu.au (L.F.); 3Curtin Medical School, Curtin University, Bentley, WA 6102, Australia; daniel.xu@curtin.edu.au; 4St John of God Health Care, Midland, WA 6056, Australia; 5Curtin School of Population Health, Faculty of Health Sciences, Curtin University, Bentley, WA 6102, Australia; 6Department of Medical Education and General Practice Research, The First Affiliated Hospital, Sun Yat-sen University, Guangzhou 510275, China; 7The Medical School, The University of Western Australia, Perth, WA 6009, Australia; 8Geriatric and Rehabilitation Medicine, Royal Perth Hospital, Perth, WA 6000, Australia; 9Institute for Health Research, University of Notre Dame, Fremantle, WA 6160, Australia; max.bulsara@nd.edu.au; 10Academic and Research Collaborative in Health and Care Economy Research Institute, La Trobe University, Melbourne, VIC 3086, Australia; m.morris@latrobe.edu.au; 11Victorian Rehabilitation Centre, Glen Waverley, Melbourne, VIC 3150, Australia; 12Ngangk Yira Institute for Change, Murdoch University, Murdoch, WA 6150, Australia

**Keywords:** chronic disease, exercise, physical activity, older adults, feasibility, barriers, adherence

## Abstract

The rising prevalence of chronic diseases could be mitigated by expanding community programs. This study aimed to evaluate the feasibility of delivering a community wellness program for older adults living with chronic disease. A two-group randomized controlled study, with blinded assessments, enrolling adults (≥50 years) with chronic disease, was conducted at a Western Australian community hub. Participants randomly allocated to the intervention participated in exercise groups and a wellness activity twice a week. Both the intervention and control groups received a Fitbit™ and workbook. The primary outcomes were recruitment, retention, acceptability, and suitability. The secondary outcomes measured at baseline and 12 weeks included physical function and physical activity (step count). There were 126 older adults approached, of whom 22 (17.5%) were recruited. Eighteen participants (mean age = 70.8 ± 8.1, n = 8 intervention, n = 10 control) completed 12 weeks. Two intervention participants (25% adherence) completed over 70% of sessions and eight participants (44.4% retention) accepted an ongoing 3-month program. Health problems (30.2%) were barriers to both recruitment and adherence. There were no significant between-group differences in physical function. Physical activity was significantly higher in the intervention group compared to the control group (*p* = 0.01). Tailored programs with ongoing support may be required to improve the health of older adults living with chronic disease.

## 1. Introduction

Throughout the Asia and Pacific region, 13.6% of the population is aged 60 years or over but this proportion is predicted to increase to 25% by 2050 [1]. Australians are living longer; however, the years lived with chronic disease are also continuing to rise, with concomitant increases in demand on the care system and related services [2]. Between 2003 and 2023, there was a 6.3% increase in chronic disease burden, with cancer, mental health conditions and substance abuse disorders, musculoskeletal conditions, cardiovascular diseases, and neurological conditions causing the most burden [3]. Globally, these chronic diseases kill 41 million people annually, equivalent to 74% of all deaths [4]. Over 17 million people die from chronic disease before 70 years of age and cardiovascular diseases, cancers, chronic respiratory diseases, and diabetes account for over 80% of these premature chronic disease deaths [4].

In addition to improving function, regular exercise improves primary and secondary prevention of chronic diseases, such as cardiovascular disease and diabetes [5]. Regular physical activity and exercise are critical for good health and wellbeing, and national guidelines recommend 30 min of moderate-intensity physical activity per day and weekly strength and balance training for older adults [6,7,8]. Physical inactivity is one of the principal preventable causes of chronic diseases and premature death; however, up to 50% of older adults do not meet the physical activity guidelines [5,9]. Surveys worldwide identify that at least one out of three older adults 50 years or over with at least one chronic disease are classified as inactive [5,10].

Recent evidence demonstrates that wellness activities should begin at an earlier age to slow or reverse poor health outcomes in older age [6]. Further, there is a critical window of opportunity in midlife populations to delay the onset of disability and its progression in older life by managing chronic conditions using targeted programs [11]. Therefore, older adults living with chronic disease need to be empowered from mid-life and beyond to reduce their reliance on the health system and to actively engage in their own healthy aging to reduce the trajectory of decline [5,10]. Implementing preventive healthy aging programs in the community to encourage aging in place can provide an integrated approach to care that avoids hospital-centric disease-orientated models [12,13]. Community hubs offer preventive holistic health programs and could be a source of referral for health professionals (such as doctors, allied health, and nurse practitioners) to improve functional ability and social connectedness. The community hub wellness model focuses on proactive self-management of wellness, unlike the traditional medical model where good health is defined by the absence of disease [12]. Community hubs show promise for reducing social isolation and improving physical function, but they need further evaluation [14,15].

We recently designed a novel community hub-based wellness program (CONNECT 60+) to address healthy aging during the COVID-19 pandemic [15]. A program evaluation [15,16] showed that older adults strongly supported the program and demonstrated improvements in physical activity and social connections. Following this evaluation, we developed a new program that encouraged adults (aged 50 years or over) living with chronic disease to utilize the community hub to increase their physical activity and social engagement. This new program (CONNECT 50+) was based on the original design namely, a targeted exercise program, providing evidence-based exercise interventions, coaching, and health education focusing on promoting wellness and community engagement [15,17]. However, there is evidence that people with chronic disease may find it difficult to adhere to exercise interventions [18,19,20]. A systematic review found that patients living with chronic disease may require extra support to adhere to home-based exercise [18]. There is also limited evidence about how to measure and promote adherence when developing physical activity programs for people living with chronic disease [19]. Therefore, we identified the need to conduct a preliminary study to determine whether it would be feasible to conduct an effectiveness trial of the wellness program within this population.

The primary aim of the study was to assess the feasibility of conducting a large randomized controlled trial (RCT) to evaluate the effectiveness of a twelve-week community wellness program (CONNECT 50+) on the physical function of people aged 50 years and over with a chronic disease, following hospitalization, with smart watch activity monitoring and an activity workbook compared to smart watch activity monitoring and activity workbook alone. The secondary aim was to evaluate the effectiveness of the CONNECT 50+ program on participants’ physical function, levels of physical activity, and health-related quality of life.

## 2. Methods

### 2.1. Study Design

A two-group feasibility randomized controlled trial was conducted with a primary focus on feasibility outcomes. Participants were enrolled between 2 May 2023 and 30 January 2024. The trial was registered with the Australian New Zealand clinical trials registry (registration number: ACTRN12623000228684) and followed the Consolidated Standards of Reporting Trials (CONSORT) guidelines, with extension for feasibility trials (see Appendix A) [21]. Feasibility trials are important to conduct prior to implementing and evaluating interventions in larger trials [22]. Measuring feasibility can assist in identifying if the intervention should be prioritized for further testing and can also identify if the protocol needs modifications prior to implementing larger-scale programs [22]. For the present trial, we rationalized that while exercise for people living with chronic disease has established effectiveness, it has not been evaluated by delivering it as part of a self-directed wellness program in a community hub. Community hubs are primarily designed to encourage a broad focus on healthy aging for older adults rather than addressing tailored programs for older adults living with chronic disease [23]. This study used a previously described framework that conceptualizes evaluating feasibility by examining acceptability (how the intended individual recipients react to the intervention), demand (assessed by gathering data on and documenting the use of selected intervention activities in a defined population and setting), implementation (discusses the ability for an intervention to be implemented as planned), practicality (constraints/barriers found when delivering the intervention), adaptation (modifications required to accommodate the context and requirements of a different population), integration (the level of change required to integrate a new program or process into a pre-existing program), and expansion (the potential for a program to be successful within a different setting) [22,24].

### 2.2. Setting

CONNECT 50+ was delivered at a community hub in metropolitan Perth, Western Australia. Community hubs focus on specific populations and aim to provide multipurpose services that meet the needs of the local community. The participating community hub was established 60 years ago and, in 2018, became an official hub that focused on active aging [23]. The hub comprises a strong base of contributing members and volunteers who live in the surrounding area. Members pay an annual fee to join the hub and then pay a fee to participate in various activities and events. Participants were recruited from a rehabilitation clinic at a hospital in the Perth metropolitan area or a general practitioner (GP) clinic, both located within approximately five kilometers of the community hub. It was deemed that older adults with chronic disease attending these locations would be likely to live in the vicinity of the community hub, as participants were required to provide their own transport. 

### 2.3. Participants

Inclusion criteria were as follows: being aged ≥ 50 years; living in the community in their own home, with at least one chronic health condition confirmed by a treating GP; following a hospital inpatient admission or attendance at a hospital outpatient clinic within the past 3 months; able to independently ambulate with or without an aid. Potential participants were excluded if they were not medically able to complete an exercise program as determined by their GP, were unable to independently transfer from a chair to a standing position, or were unable to walk independently with or without a walking aid. Participants with impaired cognitive capacity (score less than 7/10 using Abbreviated Mental Test Score [25]) or limited English language proficiency who were unable to understand the instructions and provide informed consent were excluded from the trial.

### 2.4. Randomization and Blinding

Randomization (1:1 basis to either CONNECT 50+ intervention or to control group) took place after completion of baseline measurements. A computer program automatically generated an allocation sequence. Entry of baseline measurements online triggered the allocation to groups, which was conducted through REDCap (a secure online database management service) and overseen by the program manager, who was not involved in recruitment or data collection. Participants were informed of their allocation to a group by a research assistant who monitored participants’ attendance and trial procedure at the community hub. Baseline and follow-up measurements were conducted by an outcome assessor who was blinded to group allocation. Participants could not be blinded to their group allocation but were reminded not to reveal their allocation to the outcome assessor.

### 2.5. Intervention

The CONNECT 50+ Program was based on the CONNECT 60+ program [15] and was delivered in addition to usual care. CONNECT 50+ aimed to address physical, social, occupational, spiritual, intellectual, environmental, and psychological domains of wellness [12]. The program comprised an evidence-based strength and balance exercise class, supervised by a qualified, trained instructor, with two 45 min sessions a week over twelve weeks. Based on their personal interests, participants also completed a weekly wellness activity selected from a schedule of activities offered at the hub. Available activities included Tai Chi, Pilates, meditation, chair yoga, dancing, creative writing, art, choir, and book club. Each participant received a workbook (Healthy Aging for Midlife and Beyond, (see Appendix A) that outlined the seven domains of wellness (physical, social, intellectual, emotional, environmental, vocational, and spiritual) [26], self-management of health, principles and benefits of healthy aging and wellness, and cues to action. Intervention group participants were informed that they could undertake a free 3-month program at the community hub at the conclusion of the 12 weeks of the trial. They also received a Fitbit™ (*Fitbit*™ *Inspire 2*) with a free, publicly available app designed to log physical activity (daily step count) via a website or smartphone app (android and iOS compatible). Their step count was monitored over the 12-week trial period by a weekly telephone call from the blinded outcome assessor. The assessor asked participants to read the weekly step count from their phone and recorded the observation in the online database. If the participant did not have a smart phone or access to the internet via a home email, a pedometer (*LA Gear or iVOLE*) was used. Participants were instructed in its use and the same monitoring procedure was conducted.

### 2.6. Control Group

Participants in the control group received a Healthy Aging for Midlife and Beyond workbook and a Fitbit™ or pedometer. Their step count was monitored weekly using the same procedure. Control group participants were asked not to attend the community hub for 12 weeks and were informed they could undertake a free 3-month program at the community hub at the conclusion of the 12 weeks of the trial, with choice of the same range of programs as offered to the intervention group. All participants in both groups were instructed to continue with their usual physical and wellness activities.

### 2.7. Outcomes

The primary outcome was an evaluation of the feasibility of evaluating the effectiveness of CONNECT 50+ program in a larger trial. Feasibility outcomes, measured using both quantitative and qualitative data, were as follows:Recruitment rate by examining the proportion of screened patients converted to recruitment through the hospital and GP practice; time taken to recruit participants through each recruitment route; barriers to recruitment (feasibility of 6 months to recruit sample).Retention rates, flow of participants through the trial, barriers to participation, and adherence to the intervention (feasibility for 60% adherence).Acceptability and suitability of the intervention and trial procedures.

Qualitative feedback pertaining to the study feasibility, including reason for enrollment, response to the workbook and Fitbit™, feedback about the program if in the intervention group, any changes noticed from participation in the program, and barriers and enablers to engaging in the program, was gathered from participants in both groups at the completion of their participation in the program. The research assistant conducted a telephone interview using a semi-structured interview guide (see Appendix A). The interviews ranged in length from 10 to 45 min and were recorded and transcribed. Feedback about the study feasibility (including barriers and enablers to recruitment, engagement, and implementation) was also gathered from a hub staff member who was involved in welcoming the participant to the program and providing orientation to program activities and two research staff at the trial conclusion through an online survey (see Appendix A). Other data relating to feasibility were collected by the research assistant who completed a detailed, weekly procedural diary throughout the trial regarding participant feedback, hub observations, and any barriers observed regarding recruitment and retention.

Secondary outcomes (described in full in Appendix A) were measured at baseline and 12-week follow-up. These outcomes were physical function, measured using the Timed Up and Go (TUG) test, where less time taken indicates better physical function [27,28], and gait speed [29,30], where faster speed indicates better physical function. Physical activity levels were measured by daily step count using a Fitbit™ or pedometer, where more steps equal higher levels of physical activity. Health-related quality of life (HRQOL) was measured using the Health Questionnaire EQ-5D-5L, where a score of one indicates excellent HRQOL and zero indicates worst possible HRQOL [31]. A global health rating was measured using the Visual Analog Scale EQ-VAS [31], where two end points range from zero (worst health) to 100 (best health). Perceived wellness was measured using the perceived wellness survey [26]. Demographic data collected included age, gender, functional mobility, history of falls, diagnoses of chronic disease (provided by the GP in the medical clearance form), hospital admissions, and employment. History of exercise and physical activity (including home exercises, walking, and attendance at group or gym) was recorded.

### 2.8. Procedure

After providing written informed consent and receiving GP clearance, an appointment was made for participants to attend the community hub where the blinded outcome assessor administered baseline measurements and supplied participants with their workbook and a Fitbit™ or pedometer and instruction for its use. Participants were instructed that the outcome assessor would contact them weekly for 12 weeks to obtain their summary physical activity data (step count). Participants who were randomized to the intervention group attended the community hub to receive the intervention, and all participants in both groups received a weekly telephone call to monitor their physical activity. At completion of the 12 weeks, all participants attended the community hub again for follow-up measurement by the outcome assessor and returned their Fitbit™/pedometer. All recorded data were entered into a secure online database using REDCap electronic data capture tools hosted at the University of Western Australia [32]. All study procedures, including the data management plan for privacy and security, adhered to Human Research Ethics Committee requirements and the National Statement on Ethical Conduct in Human Research [33].

### 2.9. Statistical Analysis

#### 2.9.1. Feasibility Outcomes

Quantitative feasibility data were summarized using descriptive statistics, including the number and proportion of participants approached to participate who did/did not commence the program and the number of participants who completed the intervention or dropped out. Participants’ adherence to the intervention, namely weekly frequency and type of group sessions attended at the hub, was summarized using descriptive statistics. Interview transcripts of both participant and staff feedback questionnaires and observational data from the procedural diary were analyzed using deductive content analysis [34]. All findings were subsequently synthesized by applying the framework described by Bowen et al. (acceptability, demand, implementation, practicality, adaptation, integration, and expansion) [22,24] to present the final results.

#### 2.9.2. Secondary Outcomes

Demographic characteristics and secondary outcomes were summarized using descriptive statistics, with frequency distributions for categorical data and means and standard deviations for normally distributed data or medians, interquartile ranges, and ranges for non-normally distributed continuous data. Shapiro–Wilk test was used to assess the normality of the continuous variables. Generalized linear mixed-effects models with random subject effects were used to examine secondary outcomes, including TUG, gait speed, perceived wellness, EQ-5D-5L utility value, and EQ-VAS over 2 time points (baseline and 12 weeks) and between the 2 groups. Model results were summarized using marginal mean and mean difference estimates with 95% confidence intervals. Comparison of the mean weekly step count between the two groups was analyzed using independent t-tests. Stata version 16 software (StataCorp, College Station, TX, USA) was used to complete all analyses. A *p* value < 0.05 (two-tailed) indicated statistical significance.

#### 2.9.3. Sample Size

The sample size was chosen to inform sample size calculation for a future RCT. The study aimed to determine if the target sample size of n = 60 was feasible to be recruited over a period of 6 months.

## 3. Results

### 3.1. Participant Flow Through the Study

The flow of the participants through the study (including recruitment and retention) is presented in Figure 1. There were 126 adults screened, of whom 77 (61.1%) met the eligibility criteria. Of those eligible, 55 of these adults declined to proceed further, and 22 (28.6%) participants (mean age 70.8 ± 8.1 years) were enrolled. Four participants withdrew from the study. The planned sample size (n = 60) was unable to be achieved.

Participants’ (eight men and ten women) characteristics are presented in Table 1. Seven intervention group and nine control group participants had two or more chronic diseases. Follow-up interviews were conducted with 17 (94.4%) participants. One participant was not available to be interviewed.

### 3.2. Primary (Feasibility) Outcomes

#### 3.2.1. Acceptability

Most intervention group participants provided positive reviews of the exercise components of the program, having enrolled in the study in the anticipation of benefits to their health. “*At that time, I think I needed something to monitor my health, my heart condition and everything….to get me going*” (P1). “*I thought that if I kept on going…I’ll get into a healthier shape…to help my heart…so exercise was the priority…getting fit was the priority for me*” (P2). Some participants stated that joining the program increased their motivation to exercise. *“The Connect program was good for making me realize… that I should be doing more physical activity and, and finding ways that work for me”* (P20). The range of programs offered with the combination of both exercise and wellness activities added to the appeal. *“I found the motivation to sort of get into, do some exercise. I really need something structured, that if it’s self-motivation, that doesn’t tend to happen which is quite common with a lot of people”* (P22). However, two participants reported they felt uncomfortable to participate via a group exercise format. *“I have done some group things before, but I tend to get frustrated. Partly ‘cause,…I’m not as good as other people at the physical activities”* (P20).

#### 3.2.2. Demand

The program did not attract large numbers of older adults for potential screening, even with an extension of the recruiting period. Advertising the program through the local hospital via allied health departments did not attract responses. Of the eighteen GP clinics approached in the geographical area of the hub, only three regularly maintained contact. These clinics informed several potential older adults; however, there was minimal uptake by the older adults, with only n = 4 participants enrolled.

The research assistant attended 81 hospital rehabilitation clinic sessions where older patients with chronic disease were completing rehabilitation programs (on 50 separate days). However, the class numbers were lower than expected. Patients with chronic disease did not automatically undertake rehabilitation programs and may have been discharged directly to the community. Out of 324 attendances, there were 75 different patients. Recruitment occurred over a period of approximately one year rather than the six months intended. Fourteen of the enrolled participants attended this hospital outpatient setting.

The greatest barrier to recruitment (n = 55 older adults who declined to enroll) after a potential participant was approached was health-related (41.8%). The research assistant observed that *“they said they’d love to give it a go but just aren’t well enough to do it*” (procedural notes). Health problems in the population were highlighted by the composition of the final cohort, as 16 (88.8%) participants had two or more chronic diseases. Other reasons for declining included not being not interested (23.6%), being too busy (9.1%), and work-related factors (9.1%). The research assistant, although they were an experienced recruiter who had worked on other large trials, found it difficult to interest potential participants in the program: *“I got the very distinct impression that they really weren’t interested in what I had to say right from the start”* (procedural notes). A few older adults who were attending hospital rehabilitation clinics initially agreed to participate but subsequently had their course of rehabilitation extended due to ongoing chronic health issues and were unable to enroll. This length of time between expressing initial interest and formal recruitment often resulted in the older adult eventually declining to participate. Following their hospital rehabilitation, some older adults expressed their reluctance to undertake another exercise program. *“It seems that although they might understand the need to exercise to prevent further relapse/progression of their disease, this doesn’t necessarily translate to the motivation to engage in an* [ongoing] *exercise program”* (procedural notes).

Time constraints were also mentioned as a reason for declining. Family commitments including caring for a partner, baby-sitting, and involvement in other activities were recorded as *“too busy”* (procedural notes). Some participants who returned to work were unable to enroll or attend. *“My work ended up getting a lot busier…It’d gone quiet and I got really busy and because I work for myself, I couldn’t really knock back the work”* (P16). This particularly affected participants who were receiving health care to manage their chronic disease as a return-to-work pathway.

#### 3.2.3. Implementation


**Exercise groups**


Health problems were a frequently mentioned barrier to participation in the participant’s chosen physical activity once they enrolled. Over 88% of participants had two or more chronic diseases, and 15 (83.3%) participants took three or more prescription medications. It became apparent that participants’ had difficulty in self-managing their participation levels around their level of physical capacity. Some participants felt that the exercise groups were too difficult and were not modified sufficiently, even though all exercise groups allowed for individual abilities. The strength and balance exercises groups were the most difficult for participants, and impairments in mobility and balance, shortness of breath, and aggravation of existing health conditions were the most common physical barriers reported. One participant commented on fatigue preventing them from undertaking more than one activity weekly, stating that *“I found I got very tired the next day each time I went to them... I’d feel very fatigued”* (P9). Another participant commented that “*I do have balance issues, but the sort of activities that were involved were just too hard for me…I felt at risk of falling over…I didn’t really feel comfortable…. It started off fine and then got harder and harder*” (P22). These participants were able to continue their physical activity by changing to an easier group and subsequently reported on positive aspects of their experience, such as increased levels of physical activity. *“It gave me more confidence…to walk and stretch and…build up my fitness again”* (P9) and *“I found that I was physically more active than prior to starting that particular program”* (P22).


**Fitbit™ use**


Participants’ response to the Fitbit™ was mixed. Many participants (from both intervention and control groups) needed staff assistance for the initial set-up, and several needed further support when attempting to access and provide their weekly step count to the outcome assessor. *“Fitbit*™ *usage was quite a challenge. The Fitbit*™ *app was harder because some participants couldn’t access the steps, some couldn’t find the app, and some did not have updated phones so the Fitbit*™ *would not sync with the phone”* (procedural notes). It was noted that some participants did not have a smartphone. To overcome this barrier, those participants without a smartphone (n = 4) were issued with a pedometer. A participant with a pacemaker (n = 1) who was medically advised to not use a FitBit™ was also issued with a pedometer.

Some participants reported that monitoring their step count was motivational and positively influenced their daily activity levels, with one participant commenting that “*It gave me more motivation to, to do more I guess*” (P7). Another participant noted that “*monitoring my step count seems to have, at least, made me more aware of what I’m doing/not doing* [in terms of exercise]” (P13). However, Fitbit™ “anxiety” was also noted at times. *“It stresses me out because of the pulse rate and all the rest of it…I think if anything, it gives me a bit…of anxiety, not for the steps, but because I have a medical condition. And the blood pressure goes up and down, that stress gives me anxiety…sometimes it’s better not to know, than know too much”* (P14).


**Workbook**


At baseline assessment, the Healthy Aging for Midlife and Beyond workbook was provided to all participants. Although the researchers rationalized that it would be motivational and informative to provide a hard copy resource for participants, few participants, either in the intervention or control group, recalled reading it when interviewed at 12 weeks, although one suggested that a further category of *“financial wellbeing”* (P21) should be added. Another participant stated that *“it just gave a brief outline of the different areas”* (P20).


**Social Connection**


Participants reported that the social component of the intervention was a welcome addition to the exercise component that had been emphasized by the researchers as beneficial to manage chronic disease. One participant reported that “*it makes you feel better within yourself….it’s made me think that it’s no good sitting at home even though I’m retired and doing nothing*” (P3). Another participant commented that “*I was perhaps task focused…here to do an exercise, but when people were friendly...there’s another element to it…social aspect as well as the physical aspect*” (P20). Most participants reported that everyone was friendly (both staff and other hub members) when they attended groups. *“*[It] *reinforced my belief that doing things in a group is very, very beneficial…from a social perspective...I get energized by that…it’s a social contact”* (P1).

#### 3.2.4. Practicality

The practicality of delivery was impacted by the organizational structure of the hub, participants’ health. and the intervention design. The location of the hub was considered to make it practical to attend as it was within most participants’ area. Participants were recruited from a hospital or medical setting and joined the program as an opportunity to increase their wellbeing in a local community setting. Participants found the senior focus welcoming: *“I was very impressed with the range of programs that they had at Connect”* (P20) and *“I was very much surprised…. it’s fabulous….we need to find a way to engage other councils in moving…forward, there’s such a variety of things to do that would appeal to varying capabilities and age groups…within the senior ranks…”* (P2).

The community hub is open during regular business hours (Monday to Friday 9 am to 5 pm) only. This affected the accessibility for participants who were returning to work. *“A lot of the programs were…in the middle of the day that I’d selected…And that’s…. quite disruptive…because you can’t really plan some things for mornings or afternoons”* (P22). Other barriers included difficulty parking, transport, and class availability out of business hours. *“Many had limited time available and tried to put their commitment to the program all on one day and very few were prepared to commit to three activities”* (hub staff member).

Participants’ adherence to the intervention is summarized in Table 2 (*see detailed weekly attendance in Appendix A*). Intervention group participants were originally asked to enroll in three activities per week (two exercise groups and one wellness/social activity). There was sporadic engagement in both physical and wellness activities and no participants adhered to the whole program. Only two participants undertook a wellness activity weekly for 12 weeks. Even though participants were asked to enroll in two exercise groups weekly (24 total sessions of exercise), two participants attended 20 and 25 groups, respectively, and the remainder (n = 6) attended less than one exercise group weekly.

Participants reported that they found the attendance requirements were difficult to sustain given their current health status. *“Doing two things together…I found that was a bit much”* (P22). Some social activities were physical in nature. Due to the level of difficulty of the strength and balance activities and the preferences noted by some participants, the definition of engagement in a physical activity group was extended to include Pilates and Zumba as an option.

From a social perspective, some participants found it hard to join groups where existing hub members had been attending for long periods of time. *“Most of them had been going there for quite some time. So they’d formed their connections and to get in usually takes a bit of time, with anybody”* (P14). This participant mentioned that whilst the staff were engaging and friendly, they felt that other attendees did not welcome and accept them. *“You had to try to make a conversation with people and they’ll answer you, and that’s all”* (P14). However, another participant felt the hub could be a useful way of avoiding social isolation, stating “[the hub provides] *a variety of things that would help them, especially people that are alone and worry about it...I think they have a lot to gain and benefit from it*” (P12).

#### 3.2.5. Adaptation and Integration

The adaptation and integration of the program into regular community hub participation were examined from two different perspectives: the participants and the staff. Participants’ response to the program was divergent and individually contextual. Some participants described how integrating their attendance into their current lifestyle was challenging. *“The usual problem most people have…they’re usually busy, so trying to fit something in is often the problem”* (P22).

Regarding the hub, some participants reported that they had not developed any longer-term social connections during the 12 weeks of the trial and felt this was a barrier to continuing. One participant felt that this was at least partly due to the short time frame of the study, stating that “*I wasn’t there going for long enough to be fitted into a group*” (P14). However, while participant integration into the regular hub activities was low, those who did attend regularly found it beneficial. One participant commented *“…thank you for introducing me to the hub… it’s an amazing set up….it’s lovely. I didn’t know places like that existed”* (P9). Another participant who had attended the hub for the first time stated *“I wouldn’t …hesitate to….explain to people and encourage them…. I think the concept is actually good”* (P22).

Both research staff commented that the hub was excellent and had a wide variety of activities and events offered. *“A sense of community is present…staff are very welcoming”* (research staff survey). However, research and hub staff concurred with participants that the program appeared to be better suited to people who are relatively well, not those dealing with a chronic health condition or recent acute illness. One of the research staff questioned the suitability of the program: “*Is the Connect Program suitable for a group of people with chronic disease? Few of the participants themselves seemed to think so. It seems to be better suited to those who are generally healthy to begin with. Rather than those who have chronic disease, have recently been ill enough to be in hospital…*” (research staff survey). The hub staff member also commented that *“some of the people that were recruited were unlikely to participate in the exercise activities as they obviously had physical challenges that would make participating very difficult”* (hub staff survey).

#### 3.2.6. Expansion

The potential to expand the program for people living with chronic disease appeared limited due to the substantial difficulties with recruitment and the intervention group’s low levels of adherence to the intervention. A further 3-month free membership at CONNECT was offered to all 18 participants at the completion of the trial (the 12-week follow up assessment), including the control group. Thirteen (72.2%) participants (n = 7 control group, n = 6 intervention group) were interested in this option. Eight of these participants (44.4%) accepted the free membership. However, after three months, when these participants were contacted by phone, only four (22.2%) participants had attended for more than two sessions over this period. Reasons for not taking up membership included health problems, returned to work, too tired, too busy, attending other exercise groups, language difficulty, and transport difficulties. At completion of the 3-month extended membership, two (11.1%) participants paid to continue as a hub member (x1 intervention, x1 control).

### 3.3. Secondary Outcomes

Changes in physical function and other secondary outcomes within and between the intervention and control groups are presented in Table 3. There were no significant differences between the two groups in gait speed, the TUG test (physical function), self-rated health (global health rating), the EQ-5D-5L (health-related quality of life), or perceived wellness survey (perceived wellness).

The mean weekly step count (physical activity) per week for both groups is presented in Figure 2. The intervention group demonstrated a significantly increased weekly step count compared to the control group (intervention group = mean steps 53,112.8 (±16,917.3) vs. control group = mean steps 28,855.8 (±16,312.2), *p* = 0.01). The intervention group completed an average of 568,000 (±206,953.7) total steps in 12 weeks compared to the control group who completed an average of 345,855 (±196,430) total steps (*p* = 0.03).

## 4. Discussion

Our study found significant barriers to delivering an intervention designed to facilitate older people living with chronic disease to participate in a community wellness program. The problems with recruitment and adherence indicated the intervention was not feasible to be evaluated within our chosen population. The program provided a wide choice of activities, there were no financial costs associated with participation, and all activities were delivered in participants’ local community in a setting that was specifically targeted to older adults. Despite this design, it took double the allocated time to enroll 22 of the target sample size of 60 participants after which the trial was closed due to the poor recruitment rate. Only 28% of eligible participants agreed to enroll, and only two of eight intervention group participants showed good adherence to the program (well below the 60% adherence target). Recruitment targeted adults (50 years and older) who may have recently been hospitalized, as there is evidence that wellness activities should begin at an earlier age to slow or reverse poor health outcomes in older age [6] and people living with chronic diseases need to regain daily function and re-engage with their community after hospitalization [35]. The CONNECT 50+ program design incorporated the widely accepted biopsychosocial model of health rather than a medical model of care and was based on our successfully piloted wellness program [15,16]. However, this population of adults aging with chronic disease reported significant health, time, and motivational barriers to self-managing their engagement in a selection of physical and wellness activities.

Overall, participants’ feedback indicated that for older adults with chronic disease, the practicality of attending a community hub to access exercise, health and wellness activities may be a daunting and difficult task. Almost all participants had two or more chronic diseases, and most reported an inability to adhere to the physical activities, commonly due to fatigue and difficulty completing strength and balance exercises. The physical and wellness activities were tailored in terms of type, frequency, needs, and interests; however, all activities were conducted in a group setting, which required participants to largely self-manage adjustments of intensity and duration during the activities. The intervention was delivered within a group exercise design, which is known to have positive benefits for older people, including improved adherence, social interactions, and quality of life compared to individual exercise programs [36,37]. However, hub staff were concerned that the requirement to attend at least two physical activities was beyond the capacity of some participants. This perspective was confirmed by participants’ feedback that they lacked capability to engage and adapt to the program and gain positive effects. The majority of participants attended less than one exercise session per week, which was not consistent with physical activity recommendations [6], and this was reflected in the lack of improvement in secondary outcomes either within the intervention group or between the groups, other than in step count. Lower attendance has been shown to lead to lower adherence, especially when individuals have low exercise self-efficacy [20].

Overall, these findings suggest that adults who are aging with chronic disease may require tailored programs, multidisciplinary team support, and psychological health support and education to engage in exercise programs despite their fluctuating physical ability [20]. Providing personalized support with exercise dosing has been identified as an essential skill in primary care to encourage individuals to progressively develop exercise-related self-efficacy and progress exercise dosing as exercise capacity increases [38] and is supported by national guidelines for exercise and physical activity [7,8]. There were also low levels of engagement in the wellness activities, although participants may likely have been engaging in other social activities outside the program. Social programs are an evidenced-based way to improve social connectedness and psychosocial outcomes for people living with chronic disease [39,40]. However, some participants failed to make social connections during the program, which may have improved their attendance and subsequent adherence to physical activities. Social connections are known to be an important enabler to improving engagement in exercise among for older adults with chronic disease [16,20]. Older adults who had joined an earlier program at the hub experienced a sense of social connection that was not evident in this trial [16]. This may have been because participants in our trial had chronic disease and had reduced levels of physical fitness to engage in multiple and sustained social activities. Older people with chronic disease report that physical limitations and attitudes to forming new social connections, such as not feeling confident, limited their ability to form social connections [41]. A recent scoping review found that group dynamics may act as barriers to some people with chronic disease participating in group programs [42]. Recommendations from two reviews suggest that community organizations pay particular attention to collaboration with end-users and individualized consultations, including using social and peer support to enhance enrollment and attendance [20,42].

Other aspects of the program delivered to both intervention and control groups also demonstrated mixed effects. FitBit^TM^ activity monitors were introduced to both groups as a simple intervention recognized to address the public health challenge of inactivity [43]. Building self-efficacy to monitor and manage physical capacity in real time was identified as an enabler for some participants in both groups; however, for others who became more aware of the signs and symptoms of their disease, this was a source of anxiety and a barrier to engaging in physical activity. The workbook was issued to all participants and was designed to provide simple and motivational information. However, few participants remembered reading the book or finding it useful. Participants may have required more individualized discussion to tailor the book’s messages for their gaps in knowledge, motivation, or awareness about the importance of physical and social activity for managing their chronic disease [44].

When findings are examined by applying a health behavior framework, they demonstrate that only a few participants gained sufficient motivation or capability or used the provided opportunities to undertake the desired behavior of self-managing their engagement in exercise and wellness activities [44]. Although the sample size in our study was small, the findings suggest there is a substantial gap in Australia between older adults living with chronic disease being provided with medical care and progressing to independently self-managing their disease. Senior centers, such as used in our study, offer a wide range of programs but non-attendees to centers have been reported to be unsure of their value or uninterested in attending [45]. This is a significant problem in the context of the global aging population, as our program provided evidence-based physical and social activities that are known to be critical for healthy aging [46,47]. Our findings contrast with an evaluation of an intensive home exercise program for older frail people, in which participants demonstrated significant functional improvements [48]. However, that trial enrolled participants with support from a hospital setting and provided intensive health professional coaching and home visits. We aimed to encourage older adults with chronic disease to engage in self-managed activities in the community to avoid a hospital-centric model of care [12,13]. A further review of interventions designed for community management of chronic disease are warranted, including asking clinicians who treat these older adults to empower their patients to take care of their own health, as clinicians are known to be trusted authorities for health information and social prescribing by the target population [49]. Social prescribing using link workers has been suggested as an implementation strategy to enable the transition between hospital and primary health services by using consistent methods of gradually scaling coaching and empowerment of patients to understand the effects of their disease on the capacity to exercise and engage in other wellness activities. These link workers could also provide support to prioritize external factors such as health appointments and family commitments [50]. National guidelines provide information for older adults and health workers that can aid the implementation of healthy aging strategies and support these kind of social prescribing programs [8,12,51,52].

### Strengths and Limitations

The strengths of this trial were the blinded outcome assessors, the randomization procedure that was managed independently through an online service, and the detailed feedback pertaining to feasibility that was provided by participants and staff. The feasibility design allowed the team to identify that it would not be appropriate or practical to evaluate the efficacy of the intervention in its current design in a large RCT in this population or setting. There were important limitations. We only recruited within two organizations and recruitment might have been improved with further sites. The trial design focused on evaluating self-management and group interactions through attending a community hub, precluding us from exploring further practical steps, such as offering more health professional support, which might have improved adherence. The final sample size was very small, and the trial was only conducted in only one community. Of necessity, individual activities in the hub are run by different instructors who are experts in their field, meaning that intervention fidelity may have varied within the intervention group. We enrolled participants with any chronic disease and hence findings may be less applicable for specific diseases. The generalizability of the findings depends on other community program settings, as this type of intervention may be more feasible if additional tailored support was provided. Our setting was an independent community hub where older adults initiate and participate in a wide range of activities that are self-selected.

## 5. Conclusions

Population aging is reported to be the most important medical and social demographic problem globally [47]. A personalized wellness program including physical and social activities for adults aged 50 years and over living with chronic disease was not feasible to implement in a general community hub. Health conditions impacted interest and availability for enrollment and program adherence. Program-level barriers identified that older adults living with chronic disease may require support from health professionals to initially develop, engage in, and adhere to a self-managed exercise and activity program. Health care workers play an important role in encouraging older adults living with chronic disease to self-manage their health. Therefore, providing education, referrals, and motivational approaches to encourage these older adults to be physically active is important to prioritize when treating this cohort. Health care providers may need to offer more sustained support, including individualized health worker support, to assist older adults living with chronic disease to transition to community programs as a means of independently maintaining and improving their health. Strengthening partnerships between community organizations and health care providers could provide more opportunity for structured referrals and consequent engagement in ongoing exercise and wellness activities. While the program design incorporated the biopsychosocial model of health, the complexity of incorporating these elements at an individual level remains. Further research is warranted to consider how to develop programs that assist older adults living with chronic disease to develop the capability, opportunity, and motivation to change their behavior to self-manage their physical and wellness activities and to promote healthy aging in this population.

## Figures and Tables

**Figure 1 ijerph-21-01667-f001:**
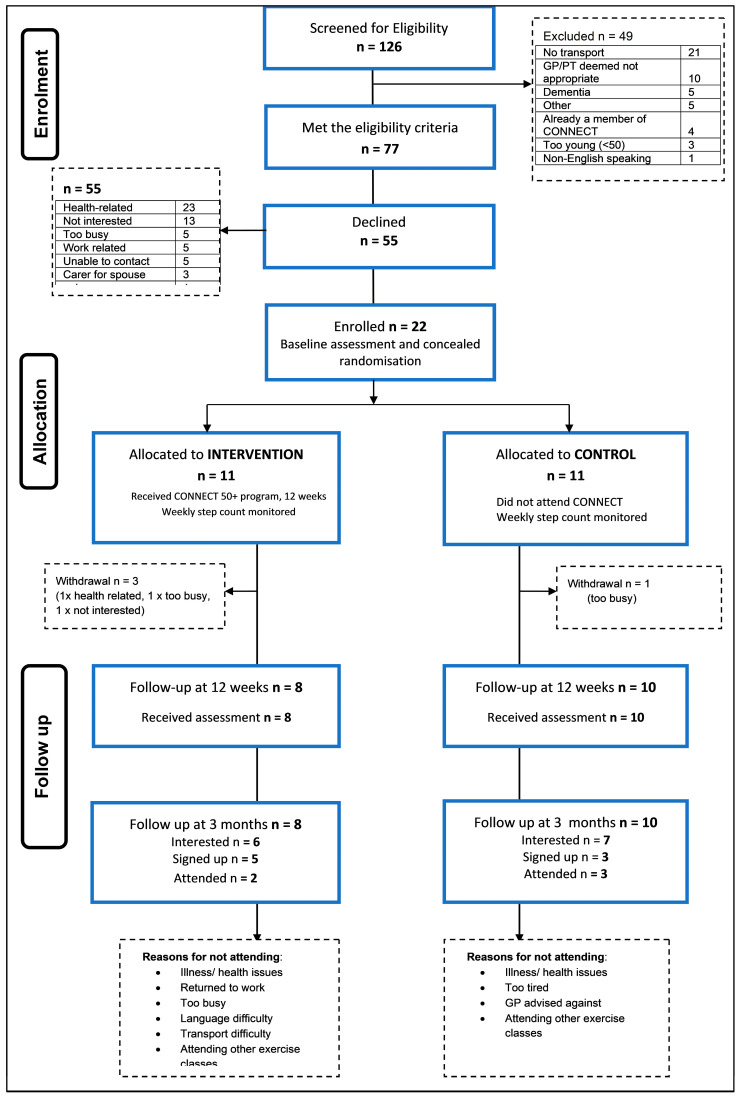
Participant flow through the study.

**Figure 2 ijerph-21-01667-f002:**
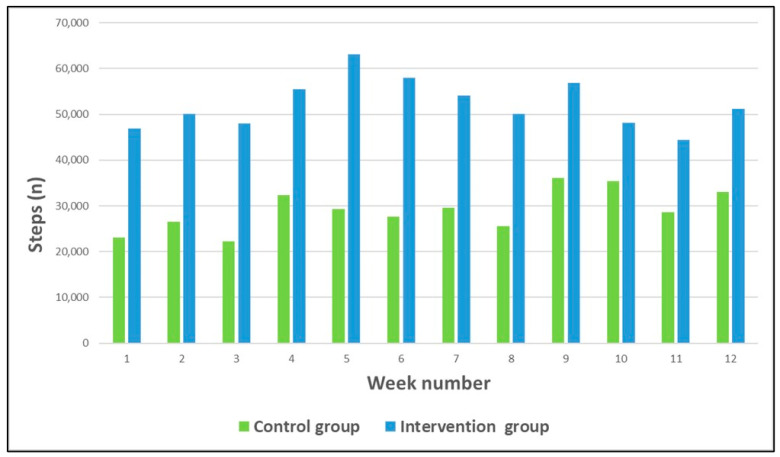
Mean weekly step count for intervention and control groups.

**Table 1 ijerph-21-01667-t001:** Participant characteristics.

Characteristic	Intervention n = 8	Control n = 10
Age (Years), mean (SD)	69.6 (9.2)	71.7 (7.5)
Sex (n; %)		
Men	2 (25.0)	6 (60.0)
Women	6 (75.0)	4 (40.0)
Employment ^a^ (n; %)	2 (25.0)	3 (30.0)
Fell in last 12 months (n; %)	3 (37.5)	3 (30.0)
Walking aid use ^b^	2 (25.0)	1 (10.0)
Exercise activities ^c^ (n; %)		
Nil	1 (12.5)	2 (20.0)
1–5 x/week	2 (25.0)	4 (40.0)
Daily	5 (62.5)	3 (30.0)
Walking (n; %)		
Nil	1 (12.5)	3 (30.0)
1–4 x/week	2 (25.0)	4 (40.0)
Daily	5 (62.5)	3 (30.0)
Home care services ^d^ (n; %)	3 (37.5)	1 (10.0)
Lifestyle risk factors identified by doctor (n; %)		
Smoking	1 (12.5)	0 (0)
Alcohol use	0 (0)	2 (20)
Low physical activity levels	2 (25.0)	3 (30)
Sleep problems	1 (12.5)	1 (10)
Nil	6 (75.0)	6 (60)
Number of chronic diseases (n; %)		
1	1 (12.5)	1 (10.0)
2	4 (50.0)	2 (20.0)
>2	3 (37.5)	7 (70.0)
Chronic diseases (n; %)		
Respiratory	2 (25.0)	4 (40.0)
Cardiovascular	6 (75.0)	9 (90)
Renal	0	2 (20.0)
Diabetes	1 (12.5)	4 (40.0)
Gastrointestinal ^e^	1 (12.5)	3 (30.0)
Arthritis/musculoskeletal	5 (62.5)	4 (40.0)
Depression/Anxiety	1 (12.5)	1 (10.0)
Other ^f^	3 (37.5)	7 (70.0)
Prescription medication (n; %)		
0–2	2 (25.0)	1 (10.0)
3–5	4 (50.0)	4 (40.0)
≥6	2 (25.0)	5 (50.0)
Hospital Admission (n, %)		
Inpatient in previous 3 months	5 (62.5)	6 (60)
Attending hospital for outpatient rehabilitation in previous 3 months	7 (87.5)	7 (70.0)

*Table notes*: a = paid, part-time, or full-time employment; b = walking stick or walker; c = includes attending group exercises and/or home exercises; d = includes home care, transport, personal care; e = diverticular disease, GORD, hemicolectomy, liver problem; f = sleep apnea, retinopathy, COVID-19, allergic rhinitis, cerebral atrophy, parkinsonism, cancer, dermatitis.

**Table 2 ijerph-21-01667-t002:** Participants’ adherence to the program.

Participant (n = 8)	Physical Activity Group, n	Wellness Activities, n	Total Activities Completed n = 36, (100%)	Adherence (%)
a	20 ^a^	12 ^b^	32	88.9
b	7 ^a^	5 ^c^	12	33.3
c	3 ^a^	7 ^c,d,e^	10	27.8
d	9 ^f,g^	2 ^c^	11	30.6
e ^i^	0	0	0	0
f	5 ^a^	0	5	13.9
g	10 ^a,g^	13 ^b,h^	23	63.9
h	25 ^a,g^	1 ^c^	26	72.2

*Table notes*: a = strength and balance, b = Tai Chi, c = chair yoga, d = community lunch, e = independence and wellness group, f = Zumba, g = Pilates, h = writer’s group, i = participant returned to full time work.

**Table 3 ijerph-21-01667-t003:** Secondary outcomes.

	Time	Intervention Mean (95% CI)	Change from Baseline (Within Group) Mean (95% CI)	*p* #	Control Mean (95% CI)	Change from Baseline (Within Group) Mean (95% CI)	*p* #	Difference Between Groups Mean (95% CI)	*p*	*p* *
**Timed up and go (s)**	B	9.84 (7.76, 11.92)			12.15 (10.28, 14.00)			−2.31 (−5.10, 0.49)	0.09	
	12 w	10.16 (8.07, 12.24)	0.32 (−0.87, 1.51)	*0.60*	11.87 (10.01, 13.74)	−0.27 (−1.34, 0.80)	*0.62*	0.59 (−1.01, 2.19)	*0.50*	*0.44*
**Gait speed (m/s)**	B	1.17 (0.98, 1.36)			1.29 (1.12, 1.46)			−0.12 (−0.37, 0.13)	*0.34*	
	12 w	1.20 (1.01, 1.39)	0.03 (−0.08, 0.14)	*0.60*	1.27 (1.11, 1.44)	−0.01 (−0.12, 0.09)	*0.76*	−0.08 (−0.32, 0.17)	*0.55*	*0.55*
**Wellness (score)**	B	8.37 (6.58, 10.17)			9.44 (7.83, 11.05)			−0.55 (−2.96, 1.85)	*0.65*	
	12 w	8.28 (6.48, 10.07)	−0.10 (−1.33, 1.14)	0.88	8.83 (7.23, 10.44)	−0.61 (−1.71, 0.50)	0.28	0.51 (−1.50, 2.17)	0.55	0.55
**EQ-5D-5L utility value**	B	0.88 (0.76, 1.02)			0.78 (0.67, 0.90)			0.10 (−0.07, 0.27)	0.26	
	12 w	0.86 (0.73, 0.99)	−0.07 (−0.12, 0.64)	0.57	0.74 (0.62, 0.86)	−0.04 (−0.12, 0.04)	0.32	0.11 (−0.05, 0.29)	0.19	0.81
**EQ-VAS**	B	70.00 (55.88, 84.12)			73.30 (60.67, 85.93)			−3.30 (−22.24, 15.64)	0.73	
	12 w	74.37 (60.26, 88.49)	4.37 (−8.06, 16.81)	0.49	67.50 (54.87, 80.13)	−5.80 (−16.92, 5.32)	0.31	6.87 (−12.06, 25.82)	0.48	0.23

*Abbreviations:* B = baseline; w = weeks; CI = confidence interval; m/s = meters/second; *p* # = within group change from baseline; *p* = cross-sectional mean difference; *p* * = interaction effect (rate of change difference between groups).

## Data Availability

The original contributions presented in the study are included in the article/Appendix A; further inquiries can be directed to the corresponding author.

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
