# Peer review of "Promoting Healthy Aging for Older People Living with Chronic Disease by Implementing Community Health Programs: A Randomized Controlled Feasibility Study"

_ijerph, 2024, doi:10.3390/ijerph21121667_

Round 1
Reviewer 1 Report
Comments and Suggestions for Authors
This research offers insight into a new type of exercise program for seniors.
Thank you for this research. I appreciate the qualitative aspect and the quantitative perspective. There are a few additions I would suggest. There was such a lack of response, number 298, I would add more about why do you think this happened. I think you could add to the literature information from Go4Life from the National institute for aging, usa. The failure of many to make a social connection is a huge finding which seems to be overlooked, number 433. Add more on this. There is missing information in your conclusion, as well as suggestions for health care workers, and the application of your paper for others. Please add more on this information.
Author Response
Reviewer 1.
Thank you for this research. I appreciate the qualitative aspect and the quantitative perspective.
Response: Thank you for your helpful review and the feedback to improve the presentation of the study.
Please see detailed response below to comments. Note we have inserted new references in the reference list where referred to in the responses.
- There was such a lack of response, number 298, I would add more about why do you think this happened.
Response: We do agree with the reviewer about the lack of response and were very disappointed that the idea of using a community hub to address this cohort’s issues was not feasible. We have nearly 500 words of results in the section to describe this point so feel any more in the results would be repetition. In the discussion, we do state that it was “daunting and difficult” and we are not sure how to state this any more strongly. We are mindful that the limitations of a feasibility study, as mentioned in our limitations section, may mean that local factors were impacting on our study but we have described these as fully and directly as possible.
However, we have expanded our discussion responding to reviewer feedback by using a suggestion of reviewer 2 (see response to Reviewer2 comment number 4) to contrast our study with a study that provided more support and noted improvements. Also, we expanded the point made by stating another factor which we now describe (from responding to the useful comment number 4 below). We now discuss that health care providers need to pay more attention to supporting older people with chronic disease to move to community programs. We state in the conclusion “Health care providers may need to offer more sustained support…”) and we provide a new and recent reference about co-design (reference number 42, Kessler et al) which we think is timely and relates to this significant problem.
See the response to comment 4 below and the new reference inserted. See also the response to reviewer 2, comment number 4.
- I think you could add to the literature information from Go4Life from the National institute for aging, usa
Response: Agree, this would be a helpful reference of high quality. We decided to add this into the discussion to point out relevant evidence-based programs that readers could access. We added the latest website for the reader as the go4life page (https://www.nia.nih.gov/research/blog/2014/07/ go4life-nia-health-education-campaign) states that it has closed and to use the new webpage: https://www.nia.nih.gov/health/exercise-and-physical-activity. In the discussion we now add this reference and one other linked reference (https://www.nia.nih.gov/health/health-care-professionals-information ) and text that suggest that this evidence can aid implementation. This information also pertains to the response to comment 4 below, where we now make more mention of application of our findings into policy and practice. See discussion paragraph 5 where we have inserted highlighted text. As part of this response, note we deleted the last sentence of paragraph 5 as we now discuss social connections more deeply in paragraph 3 so this sentence was repetitive.
“National guidelines provide information for older adults and health workers that can aid implementation of healthy ageing strategies, as well as support these kind of social prescribing programs (new references 51, 52).”
- The failure of many to make a social connection is a huge finding which seems to be overlooked, number 433. Add more on this.
Response: Agree. We inadvertently did not discuss this aspect fully. We have now addressed this in the discussion and provided two new, recent references for this point which can inform readers. See discussion paragraph 3, where we now added the following text. See highlighted inserted text.
“However, some participants failed to make social connections during the program which may have improved their attendance and subsequent adherence to physical activities. Social connections are known to be an important enabler to improving engagement in exercise among for older adults with chronic disease (16, 20 ). Older adults who had joined an earlier program at the hub experienced a sense of social connection that was not evident in this trial. This may have been because participants in our trial had chronic disease and had reduced levels of physical fitness to engage in multiple and sustained social activities. Older people with chronic disease report that physical limitations and attitudes to forming new social connections, such as not feeling confident, limited their ability to form social connections (reference 41 Thompson et al). A recent scoping review found that group dynamics may act as barriers to some people with chronic disease participating in group programs (Kessler et al). Recommendations from two reviews suggest that community organisations pay particular attention to collaboration with end-users and individualised consultations, including using social and peer support to enhance enrolment and attendance (reference 42 Kessler et al).”
- There is missing information in your conclusion, as well as suggestions for health care workers, and the application of your paper for others. Please add more on this information.
Response: Agree – an excellent point. We have added more details in the conclusion about how lessons learned from this study apply to health care workers and organisations. Text has been added as below in the conclusion section. See inserted text as below in the conclusion.
“Health care workers play an important role in encouraging older adults living with chronic disease to self-manage their health. Therefore, providing education, referrals and motivational approaches to encourage these older adults to be physically active is important to prioritise when treating this cohort. Health care providers may need to offer more sustained support, including individualised health worker support, to assist older adults living with chronic disease to transition to community programs as a means of independently maintaining and improving their health. Strengthening partnerships between community organisations and health care providers could provide more opportunity for structured referrals and consequent engagement in ongoing exercise and wellness activities.”
Reviewer 2 Report
Comments and Suggestions for Authors
Thank you for allowing the reviewer to assess this manuscript. The reviewer has offered the subsequent recommendations and observations.
· The author has summarized the study clearly and understandably in the abstract.
· The reviewer contends that the authors' introduction has addressed significant issues but should include that this study aimed to evaluate the feasibility of a service program. The reviewer contends that the primary outcome should be the level of satisfaction or dissatisfaction with the exercise program.
· The reviewer suggests that the authors have presented the study results in both quantitative and qualitative formats. Do the authors classify this research as a mixed-methods study?
· What measures do the authors implement to avert contamination between the experimental and control groups among volunteers in the same area?
· The reviewer concurs with the authors' discussion but wishes to incorporate additional points regarding the characteristics of exercise programs designed for specific fitness groups, particularly those with robust, pre-frailty, frailty conditions, such as the Vivifrial program.
Author Response
The author has summarized the study clearly and understandably in the abstract.
Response: Thank you for your positive feedback and your helpful reflections to improve the final presentation of the results of the study.
- The reviewer contends that the authors' introduction has addressed significant issues but should include that this study aimed to evaluate the feasibility of a service program. The reviewer contends that the primary outcome should be the level of satisfaction or dissatisfaction with the exercise program.
Response: We agree that levels of satisfaction about the program was an important point in what we found and participants not liking some aspects informed the feasibility of continuing to refer people with chronic disease to the program. However while this will certainly inform future research, we haven’t been able to make any changes in the methods. We chose the primary aim a priori - which was to evaluate the feasibility of conducting a larger randomized trial on the same intervention. We also decided on the primary feasibility outcomes a priori based on evidence from other feasibility trials. Since the study design was in our registered trial protocol (on ANZ clinical trial registry) and this was the ethically approved version of the methods we needed to follow the protocol exactly as registered, and report following the Consort guidelines .
- The reviewer suggests that the authors have presented the study results in both quantitative and qualitative formats. Do the authors classify this research as a mixed-methods study?
Response: Agree, we have used mixed methods of data collection and analysis and realise we didn’t state this clearly throughout. To clarify, we now added this information into the methods. See inserted text in section 2.7 outcomes where we now added “measuring using both quantitative and qualitative data…” In the statistical analysis plan we had already stated how we analysed both quantitative and qualitative data. Overall, the primary design was chosen as a feasibility randomised controlled trial so we follow the CONSORT checklist for feasibility RCTs to report the study.
- What measures do the authors implement to avert contamination between the experimental and control groups among volunteers in the same area?
Response: Agree to clarify. Participants in the control group did not attend the centre so we think it unlikely there was contamination, as each participant was enrolled separately. For the control group we informed them that they could access the program after 12 weeks and we provided the funding for them to enrol at 12 weeks. The centre did not report any control group participants who enrolled earlier than this point. We added a phrase about this in the methods to clarify this for readers. See Section 2.6 of the methods, inserted text that “Control group participants were asked not to attend the community hub for 12 weeks….”
- The reviewer concurs with the authors' discussion but wishes to incorporate additional points regarding the characteristics of exercise programs designed for specific fitness groups, particularly those with robust, pre-frailty, frailty conditions, such as the Vivifrial program.
Response: Agree. We added discussion about this while being mindful of discussion length. This comment is very interesting as the Vivifrail trial is essentially the opposite type of intervention and contrasts with our intervention, as participants in the Vivifrail trial were supported by the hospital and received coaching and ongoing support. We were trying to remove burden from the health system and encourage older people with chronic disease to initiate and self-manage their exercise using a community program. We suggest in our introduction that it’s not feasible for an ageing population to access outpatient programs in hospital as a long term means of maintaining functional ability. We now added this discussion point –see inserted text that we have added in to paragraph 5 of the discussion as below.
“Our findings contrast with an evaluation of an intensive home exercise program for older frail people, in which participants demonstrated significant functional improvements (Vivifrail trial ref 48). However this trial enrolled participants directly from a hospital setting and provided intensive health professional coaching and support. We aimed to encourage older adults with chronic disease to engage in self managed activities in the community to avoid a hospital-centric model of care.”
Reviewer 3 Report
Comments and Suggestions for Authors
Thank you for the opportunity to review article 3316781. It fits into the public health issue of caring for healthy aging. The article will not fill allows the final result of the study. In the reviewer's opinion, there are several issues to clarify:
1. the methodology lacks a thorough description of the study's feasibility principles;
2. how the chronic disease was documented, rules for receiving documents, medical history - secrecy clause;
3. what was the CONNECT 50+ program about and what elements were taken into account?
4. how and with what was the level of physical fitness of respondents assessed,
5. what criterion was used to compare the respondents during the activities significantly different from each other?
6. what was suggested by comparing the performance of different people in different exercises?
7. the study group can be a qualitative study, although it should focus more on the barriers to seniors taking advantage of the opportunities for activities in senior clubs.
Author Response
Reviewer 3.
Thank you for the opportunity to review article 3316781. It fits into the public health issue of caring for healthy aging. The article will not fill allows the final result of the study.
Response: Thank you for your helpful review and the feedback to improve the presentation of the results. We agree that this article fits into public health domain of caring for healthy ageing.
Please see detailed responses below to comments.
- The methodology lacks a thorough description of the study's feasibility principles;
Response: Agree. We now provide an explanation of feasibility principles related specifically to this study. See additional text has been added to each of the feasibility points that were addressed in the Methods section. (line 118) See the inserted text as below.
The study used a previously described framework that conceptualizes evaluating feasibility by examining Acceptability (how the intended individual recipients react to the intervention), Demand (assessed by gathering data on and documenting the use of selected intervention activities in a defined population and setting ), Implementation (discusses the ability for an intervention to be implemented as planned), Practicality (constraints/ barriers with delivering the intervention), Adaptation (Modifications required to accommodate the context and requirements of a different population) Integration (the level of change required to integrate a new program or process into a pre-existing program), and Expansion (the potential for a program to be successful within a different setting) [22,24].
- How the chronic disease was documented, rules for receiving documents, medical history - secrecy clause;
Response: All medical information provided by participant and through the medical clearance from their General Practitioner (family doctor). We added a phrase in the secondary outcomes section stating diagnoses of chronic disease, “provided by the general practitioner in the medical clearance”…
The data were stored in a secure database provided through The University of Western Australia using REDCap (Research Electronic Data Capture) data management services. The data management plan adhered to NHMRC guidelines and University HREC ethical and technical requirements for data storage and security. To clarify these points, the following statements were added into the methods, in the procedures section. See inserted text as below.
“All recorded data were entered into a secure online database. provided through The University of Western Australia using REDCap electronic data capture tools hosted at the University of Western Australia) [ re32]. All study procedures, including the data management plan for privacy and security, adhered to Human Research Ethics Committee requirements and the National Statement on Ethical Conduct in Human Research [ 33 ].
An additional two references were added at this point to confirm the source of ethics conduct for research in Australia and the database capability– see new reference numbers 32, 33.
- What was the CONNECT 50+ program about and what elements were taken into account?
Response: We realise that the structure of the introduction could lead to confusion about the CONNECT 50+ program. We have added detail to the introduction to clarify that we are talking about the original CONNECT 60+ program which then resulted in the development of the CONNECT 50+ program for this feasibility study. See the inserted, underlined text as below.
We recently designed a novel community hub-based wellness program to address healthy aging during the COVID pandemic (CONNECT 60+) [15]. A program evaluation [15,16] showed older adults strongly supported the program and demonstrated improvements in physical activity and social connections. Following this, we developed a new program that encouraged adults (aged 50 years or over) living with chronic disease to utilise the community hub to increase their physical activity and social engagement (CONNECT 50+). This new program was based on the original design namely, a targeted exercise program, providing evidence-based exercise interventions, coaching, and health education focusing on promoting wellness and community engagement [15,17].
- how and with what was the level of physical fitness of respondents assessed
Response: In our study, we did not define and assess physical fitness however we did measure physical activity (daily step count with a FitBit or pedometer) and functional ability (3m walk test and TUG) levels which are sub domains of physical fitness.
- What criterion was used to compare the respondents during the activities significantly different from each other?
Response: In this study we did not compare between different types of activities. The comparison was between those who engaged in any of the physical activities offered at CONNECT hub versus those in the control group who did not attend any group activities at all. There were not enough participants to be able to do any sub-group comparisons between types of activities participants attended, and we did not a priori plan for these. We used the timed up and go test as a measure of functional ability and the perceived wellness scale as a comparable measure of participants levels regardless of what activity they did.
- what was suggested by comparing the performance of different people in different exercises?
Response:. All physical activity groups were considered as a means of increasing physical exercise levels and were therefore able to be compared to those in the control group who did not attend the CONNECT hub during the 12 weeks. We provided detail on the adherence to the choices to demonstrate whether there was good adherence to the physical aspects of the program since these would be considered essential components of improving parameters of chronic disease (if we had proceeded to a larger trial). Participants had differing levels of fitness and physical capabilities as well as interests. The CONNECT hub was chosen as a venue due to the variety of both physical and wellness activities available to allow for more choice for participants. Providing choice for activities was seen as a potential benefit to participant engagement
- the study group can be a qualitative study, although it should focus more on the barriers to seniors taking advantage of the opportunities for activities in senior clubs.
Response: We are unsure of the meaning of the first part of the reviewer’s comment here. If the reviewer is suggesting that the study should be described as qualitative only, this is not possible due to it being a randomized controlled trial and the need for us to report according to our trial registry. Although there are qualitative components to the study including barriers to exercise and wellness activity uptake and participant and staff feedback, there was a quantitative component to indicate differences between the control and intervention groups in terms of physical benefits of exercise (ie gait speed, TUG and step count) and wellness (ie health and wellness scores).
We do agree with the reviewer about the larger implications, that there are overwhelming barriers particularly with significant health issues and lack of motivation, to seniors with chronic disease taking participating in activities at Community centres. We added another reference in here to discuss this. See reference number 45 and inserted text in paragraph 5 as below:
Senior centers, such as used in our study, offer a wide range of programs but non-attendees to centers have been reported to be unsure of their value or uninterested in attending [45]. This is a significant problem in the context of the global aging population, as our program provided evidence-based physical and social activities which are known to be critical for healthy aging.
Reviewer 4 Report
Comments and Suggestions for Authors
Thank you for the opportunity to review this manuscript on the important topic of promoting healthy aging for individuals with chronic health conditions through community health programming. The paper is overall well written, and the methods are clear. Some revisions are requested to increase clarity, including the emphasis of the setting for the manuscript, location of participants after hospital stay, understanding training of those leading programs, and overall shortening of the results section. Some limitations should be added for the areas in question. Therefore, I recommend a revise and resubmit for this manuscript to be considered for publication.
Abstract
Line 32- awkward wording at the start of this sentence
Introduction
Line 43- sentence awkward with a fragmented structure
Line 61- data from the USA when the first paragraph targets Australian seems misleading. Consider being very broad with your terms (i.e., worldwide) or clearly specify a region/country for the manuscript
Line 81- “planned to develop” sounds like you were not able to develop? Please rephrase.
Line 94- omit word "large" before RCT as your targeted number of 128 and final sample are not necessarily large?
Methods
Lines 117-119- nice explanation of the feasibility study rationale
Line 136- please spell out GP
Line 136- please detail “hospital patient” in more detail. Later in the results, the term rehabilitation hospital is used. Was it discharge from an acute hospital stay only or could it also include a rehabilitation hospital? Please clarify that post-acute or subacute rehabilitation was not considered as hospital stay. Or include as a limitation.
Line 164- the term “trained instructor” seems vague as a fitness instructor training may be vastly different than a book club instructor. Include this inconsistency as a limitation.
Results
Figure 1 is very helpful!
Table 1- Characteristic column 1, indent right and consider using italics for subtopics under headings within the characteristics
Line 314- did potential participants discharge to anywhere but home? Or were some potential participants discharging to post-acute rehab, family members’ homes, etc. Did any anticipate a near future transition into senior house (assisted or long term care)?
Line 337- omit bullet points in Implementation header. Use subheadings
Line 339- awkward and long sentence
Overall, the results section, even using headings, is long. Try to condense.
Discussion
-When speaking of the workbook, elaborate on future recommendations for a workbook to deliver health content. Was it effective? Would you recommend in the future based on your results?
-The start of the introduction indicates this study to be applicable to Australia. Need to mention in the discussion what these findings mean for Australia healthcare/community hubs specifically. Would help “book end” the manuscript.
-Consider expanding on your study’s methods of giving choice to participants to attend any programs on the Community Hub calendar, as opposed to attending a specific study-led program. And how this was unique but possibly another barrier to participant engagement and finishing study? Something to consider!
-Participant characteristics of “fell in last 12 months” could make for additional research inquiry as follow up study, especially possibly the influence on motivation.
-Nice use of supportive evidence throughout the discussion!
Author Response
Reviewer 4.
Thank you for the opportunity to review this manuscript on the important topic of promoting healthy aging for individuals with chronic health conditions through community health programming. The paper is overall well written, and the methods are clear. Some revisions are requested to increase clarity, including the emphasis of the setting for the manuscript, location of participants after hospital stay, understanding training of those leading programs, and overall shortening of the results section. Some limitations should be added for the areas in question. Therefore, I recommend a revise and resubmit for this manuscript to be considered for publication.
Response: Thank you. We appreciate your positive feedback about our study and your helpful review and feedback to improve the presentation of the results.
Please see detailed response below to comments.
- Abstract
- Line 32- awkward wording at the start of this sentence
“Of 126 older adults approached, 22 (17.5%) were recruited and 18 (mean age=70.8 +8.1, n=8 intervention, n=10 control) completed 12 weeks.”
Response: Agree. Sentence changed to – “There were 126 older adults who were approached of whom 22 (17.5%) were recruited. Eighteen participants (mean age=70.8 +8.1, n=8 intervention, n=10 control) completed the 12 week program.
- Introduction
- Line 43- sentence awkward with a fragmented structure
“Across Asia and the Pacific region 13.6% of the population is aged 60 years or over but this will rise to be 25% by 2050 [1].”
Response: Sentence has been amended –. Throughout the Asia and Pacific region 13.6% of the population is aged 60 years or over but this proportion is predicted to increase to 25% by 2050.
- Line 61- data from the USA when the first paragraph targets Australian seems misleading. Consider being very broad with your terms (i.e., worldwide) or clearly specify a region/country for the manuscript
Response: Agreed – sentence has been amended to more clearly identify the data sources. See inserted text which references world-wide sources.
“Surveys world-wide, show that at least one out of three older adults 50 years or over with at least one chronic disease are classified as inactive [5,10].”
- Line 81- “planned to develop” sounds like you were not able to develop? Please rephrase.
“We planned to develop a new program that encouraged adults (aged 50 years or over) living with chronic disease to utilise the community hub to increase their physical activity and social engagement.”
Response: Agree. To clarify, we changed the sentence to – “Following this evaluation, we developed a new program that encouraged…..
- Line 94- omit word "large" before RCT as your targeted number of 128 and final sample are not necessarily large?
“The primary aim of the study was to assess the feasibility of conducting a large randomized controlled trial (RCT) to evaluate the effectiveness of a twelve-week community wellness program (CONNECT 50+) on the physical function of people aged 50 years and over,”
Response: the word large in this context refers to whether the results of this small feasibility study warranted conducting large RCT which would have primary efficacy outcomes. Therefore we have left the sentence unchanged.
- Methods
- Lines 117-119- nice explanation of the feasibility study rationale
“The study used a previously described framework that conceptualises evaluating feasibility by examining Acceptability, Demand, Implementation, Practicality, Adaptation, Integration, and Expansion [22,24].”
Response: Thank you. Another reviewer has commented on this section and asked for further detail which has been added.
- Line 136- please spell out GP
“condition confirmed by a treating GP”
Response: GP = General Practitioner – has been changed in manuscript.
- Line 136- please detail “hospital patient” in more detail. Later in the results, the term rehabilitation hospital is used. Was it discharge from an acute hospital stay only or could it also include a rehabilitation hospital? Please clarify that post-acute or subacute rehabilitation was not considered as hospital stay. Or include as a limitation.
“being a hospital patient within the past 3 months”
Response: in this study, hospital patient refers broadly to either following a hospital admission or attendance at a hospital outpatient clinic. We meant to emphasise that participants had chronic disease that was leading to them needing to engage with the health system.
To clarify we changed the sentence to “ Inclusion criteria were being aged ≥ 50 years; living in the community, with at least one chronic health condition confirmed by a treating GP; following a hospital admission or attendance at a hospital outpatient rehabilitation clinic within the past 3 months; able to independently ambulate with or without an aid.
- Line 164- the term “trained instructor” seems vague as a fitness instructor training may be vastly different than a book club instructor. Include this inconsistency as a limitation.
“The program comprised an evidence-based strength and balance exercise class, supervised by a trained instructor, for two 45-minute sessions a week over twelve weeks”
Response: Thank you for your query. We agree that there are varying levels of training depending on the type of activity. We added this as a limitation to ensure there is no confusion. We have also added the word qualified as all physical activity class trainers are specifically qualified in their field of expertise. Our aim in this study was to increase engagement in both wellness and physical activity groups. See inserted word in the intervention section and in the limitations section we added the sentence “Of necessity, individual activities in the hub are run by different instructors who are expert in their field meaning that intervention fidelity may have varied within the intervention group. See inserted text.
- Results
- Figure 1 is very helpful!
Response: Thank you, we worked intensively as a team to create this figure.
- Table 1- Characteristic column 1, indent right and consider using italics for subtopics under headings within the characteristics
Response: noted and amended as suggested for easier reading. NOTE: for ease of track track changes we don’t show this as a track change as its very hard to view.
- Line 314- did potential participants discharge to anywhere but home? Or were some potential participants discharging to post-acute rehab, family members’ homes, etc. Did any anticipate a near future transition into senior house (assisted or long term care)?
Response: All potential participants were living in their own home. We realised this was not clear and added in the inclusion criteria “ living in their own home.” The study aim was to look at the feasibility of delivering a community-based wellness program. As such, one of the inclusion criteria was for participants to be living in the community. See this inserted text in the participants’ section of the methods.
- Line 337- omit bullet points in Implementation header. Use subheadings
Response: noted and amended, We don’t use track changes as it is difficult to read and understand. See no bullets and bold indented sub-headings. We made the subheadings bold but happy to change if we have misunderstood the journal style.
- Line 339- awkward and long sentence
Response: Agreed. We split the single sentence into two sentences.
Overall, the results section, even using headings, is long. Try to condense.
Response: We do acknowledge that it is long. However, we aimed to thoroughly explain the outcomes as we felt this could make the article useful for future researchers in this highly problematic area of health. What we note is that negative results are sometimes under-discussed and yet it is important to understand what didn’t work! The tables and figures are in the results section meaning it extends the text – we assume that the final version would look more succinct. Therefore we made no changes, hoping this is acceptable to the reviewer and editors.
- Discussion
- When speaking of the workbook, elaborate on future recommendations for a workbook to deliver health content. Was it effective? Would you recommend in the future based on your results?
Response: Agree. We now provide our suggestions based on the trial as we suggest that alongside the workbook participants need more discussion and personalised coaching from a coach or health professional. See inserted text in the discussion as below.
“Participants may have required more individualized discussion to tailor the book’s messages for their gaps in knowledge, motivation or awareness about the importance of physical and social activity in managing their chronic disease [44].”
- The start of the introduction indicates this study to be applicable to Australia. Need to mention in the discussion what these findings mean for Australia healthcare/community hubs specifically. Would help “book end” the manuscript.
Response: Agree. We do think most of the discussion applies to Australia (and other countries) but for one particular statement we have now added “in Australia” where we think the most significant gap is present. We then go onto to extend our discussion to global implications. See the inserted text in paragraph 5 where we begin to query the implications for our results.
- Consider expanding on your study’s methods of giving choice to participants to attend any programs on the Community Hub calendar, as opposed to attending a specific study-led program. And how this was unique but possibly another barrier to participant engagement and finishing study? Something to consider!
Response: We understand the reviewer’s point and agree that we may have had better participant engagement if the participants could have chosen their own activities. This is especially as several participants found it very difficult to participate in the physical activity groups. Our problem was that we were trying to engage people with chronic disease in physical activity and it was this lack of willingness to do physical exercise that formed the largest barrier. We think it does demonstrate that its difficult to engage people who are at low levels of fitness in exercise in community settings and more linked programs are needed as these people all have chronic disease and need to build their activity levels up.
- Participant characteristics of “fell in last 12 months” could make for additional research inquiry as follow up study, especially possibly the influence on motivation.
Response: Yes, we agree with this suggestion, there is definitely scope for future follow up assessing falls in people with chronic disease and their levels of Motivation. We think health coaching and intensive bridging support may be required if people are to remain more independent of the health system and proactive in managing their health.
- Nice use of supportive evidence throughout the discussion!
Response: Thank you for your positive comment on our research.
Round 2
Reviewer 3 Report
Comments and Suggestions for Authors The aging of the population and the coexistence of chronic diseases is a global problem.IIdentifying modifiable factors and creating programs to promote healthy aging is
tthe cornerstone of public health. The authors' additions allow the work to be published.